# Towards Structured Dynamic Sparse Pre-Training of BERT

## Abstract

Identifying algorithms for computational efficient unsupervised training of large language models is an important and active area of research. In this work, we develop and study a straightforward, dynamic always-sparse pre-training approach for BERT language modeling, which leverages periodic compression steps based on magnitude pruning followed by random parameter re-allocation. This approach enables us to achieve Pareto improvements in terms of the number of floating-point operations (FLOPs) over statically sparse and dense models across a broad spectrum of network sizes. Furthermore, we demonstrate that training remains FLOP-efficient when using coarse-grained block sparsity, making it particularly promising for efficient execution on modern hardware accelerators.

## 1 Introduction

The increasing task performance gains of large, pre-trained language models have fueled interest in computationally efficient unsupervised training (Kaplan et al., 2020). In recent years, sparsity has regained popularity as a technique for improving the computational efficiency of deep learning models (Hoefler et al., 2021). Current sparsity methods can be distinguished into approaches that impose sparsity on the weights of neural networks via *weight sparsity* (Frankle & Carbin, 2019; Gale et al., 2019; Bellec et al., 2017; Mostafa & Wang, 2019; Evci et al., 2019; Dettmers & Zettlemoyer, 2019; Mocanu et al., 2018; Jayakumar et al., 2020), or techniques that dynamically route activations to only interact with a subset of the network weights via *conditional sparsity* (Shazeer et al., 2017; Lepikhin et al., 2020; Fedus et al., 2021; Lewis et al., 2021).

In weight sparse training (Frankle & Carbin, 2019; Gale et al., 2019), the number of network parameters is reduced by imposing sparsity patterns on the network weights. As a result, weight sparse training can lead to significant savings in FLOPs, making it promising for scaling to larger network architectures for a given compute budget. One of the most promising candidates for weight sparse training is *dynamic sparsity* (DynSparse), which reduces FLOPs while only requiring training of sparse subsets of the over-parameterized network (Bellec et al., 2017; Mostafa & Wang, 2019; Evci et al., 2019; Dettmers & Zettlemoyer, 2019; Mocanu et al., 2018; Jayakumar et al., 2020; Liu et al., 2021a). In DynSparse approaches, the sparsity pattern imposed on the weights is continuously modified during training using pruning and re-allocation strategies. This evolution leads to a joint exploration of both network topology and parameters, which has been shown to outperform static sparsity baselines (Bellec et al., 2017; Mostafa & Wang, 2019; Evci et al., 2019; Dettmers & Zettlemoyer, 2019).

However, so far, the limited performance on language modeling task (Evci et al., 2019) has resulted in DynSparse training not seeing wide adoption for large-scale language modeling tasks despite recent advances (Jayakumar et al., 2020). Given the high cost and energy consumption of unsupervised training of large-scale language models (Strubell et al., 2019; Patterson et al., 2021), dynamic sparsity bears the potential to make pre-training more efficient and affordable. To this end, we adopt and investigate DynSparse training techniques (Dettmers & Zettlemoyer, 2019; Evci et al., 2019) for pre-training of BERT bidirectional language encoder (Devlin et al., 2018) based on the highly scalable Transformer architecture (Vaswani et al., 2017).

Our work achieves Pareto improvements versus the dense baseline using both structured and unstructured DynSparse training of BERT.

## 1.1 CONTRIBUTIONS

*Investigating dynamic always-sparse training for BERT pre-training.* We adapt the DynSparse training algorithm to BERT pre-training (Section 2.1). In particular, we find that gradient-based re-allocation (Evci et al., 2019) results in a collapse of the explored network parameters (Figure 11), which we mitigate through the use of random parameter re-allocation.

*Achieving scalable FLOPs efficient dynamic sparse training.* We compare dense and sparse methods for a given FLOPs budget and demonstrate both algorithmic scalability and Pareto improvement on the FLOPs scale, as shown in Figure 3.

*Adapting dynamic always-sparse training to block structures.* We extend the unstructured DynSparse training towards block-sparse structure (Section 3.2). In particular, we find that the choice of metric during block pruning has a strong influence on the task performance, as shown in Figure 7.

*Pareto improvements for structured DynSparse training* We show that the resulting structured DynSparse training of BERT without structured regularization gives Pareto improvements compared to the dense BERT baseline, as shown in Figure 1.

In the following section, we report the results of explorative experiments conducted to motivate our study of DynSparse training of BERT. The rest of the paper then concentrates on DynSparse training, with methodology discussed in Section 2 and results presented in Section 3.

## 1.2 IDENTIFYING SUITABLE ANGLE OF ATTACKS FOR SPARSE PRE-TRAINING

Sparse training of unsupervised language models is relatively under-explored, compared to sparse training of the supervised fine-tuning objective (Radiya-Dixit & Wang, 2020; Chen et al., 2020; Sanh et al., 2020). Consequently, we design two explorative experiments to assess whether DynSparse training is a suitable sparse training algorithm for pre-training of BERT.

Firstly, we analyze the importance of trainable parameters by keeping a random pattern of weights non-trainable (constant non-zero) or zero-valued throughout training. This experiment allows us to disentangle the role of 'zero' vs. 'untrainable' weights in the connectivity patterns, to shed light on the parameter dependence of BERT pre-training. Like zeroed weights, the constant weights are unable to encode new information about the task. Still, they might promote the propagation of signals or gradient-flow through the network, which has been considered a core aspect of some sparse training algorithms in vision (Evci et al., 2020; Lubana & Dick, 2021; Tessera et al., 2021). Non-zero parameters might also lead to better utilization of the remaining trainable parameters. However, as shown in Figure 1(a), we find that none of these effects plays a relevant role in the training dynamics, since the task performance of the network with sparsified weights (dashed orange line) matches the one with the same fraction of untrained weights (solid blue line). Different from vision models that are often based on convolutions, the transformer architecture contains large dense matrices and multiplicative interaction (Jayakumar et al., 2019). While zeroing parameters is not expected to affect the training dynamics, the performance remains bounded by the number of trainable parameters.

Secondly, we would like to analyze the effect of sparsification at different stages of the training process. For this purpose, we keep a random subset of the network parameters untrainable in the first half of the pre-training, before making them trainable in the second half (and vice versa). Unlike magnitude pruning, which eliminates learned information, freezing and unfreezing parameters ensures symmetry between the different phases (ignoring the linearly decaying learning rate schedule). The agreement in the task performance towards the end of training in Figure 1(b) indicates that *representation is continuously built up during training*, with no particular effect of when the sparsification is applied. This lack of preference is interesting, given that pre-training has been found to lead to a reduction of the intrinsic dimension (Li et al., 2018) with respect to downstream tasks (Aghajanyan et al., 2020). Our results suggest that sparse pre-training does not necessarily profit from an initial dense training phase. Therefore, we can distribute computation in a way that is both algorithmic and computationally beneficial. In DynSparse training, the network representation is "always-sparse", i.e. it does not rely on the representation of the underlying dense network, making the approach suited for sparse training of large language modeling architectures. Consequently, we believe that BERT pre-training is well suited for DynSparse training.

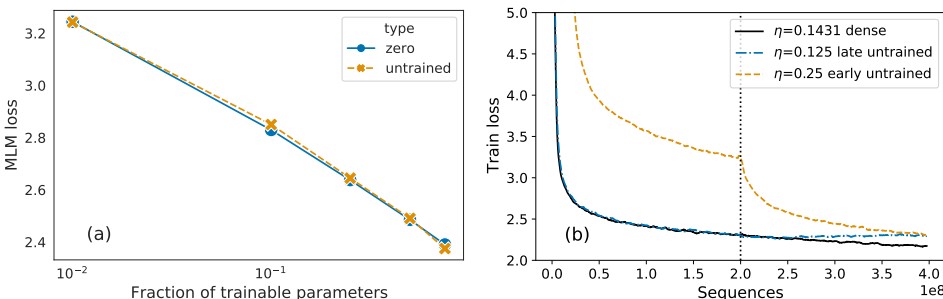

Figure 1: **(a)** MLM validation loss of BERT-Small with a random subset of parameters set to zero (solid blue curve) or kept untrained (dashed orange). **(b)** training loss curves of BERT-Small during pre-training of 10 epochs (757k steps) fixing a random subset of the parameter either early (orange dashed) or late (blue dash-dotted) during the training, as well as for the dense baseline (solid black). The vertical line indicates the unfreeze (freeze) event location, where untrainable parameters are made trainable (or trainable parameters are frozen). We pick the best learning rate for each experiment using a grid search over $0.000087 \cdot 2^m$ with $m = 0, 1, ..., 5$, given in Table A.15.

## 2 METHODOLOGY

Throughout this work, we study the self-supervised pre-training objective from the original BERT model (Devlin et al., 2018), which consists of the *Masked Language Model* (MLM) loss, corresponding to the task performance in predicting a random subset of masked tokens, and the noisier *Next Sentence Prediction* (NSP) loss for binarized next sentence prediction. We focus on a single phase of pre-training with a sequence length of 128, using the Adam optimizer. All hyperparameters are given in Appendix A for a training length of 10 epochs.

### 2.1 ADAPTING UNSTRUCTURED DYNSPARSE ALGORITHM TO BERT

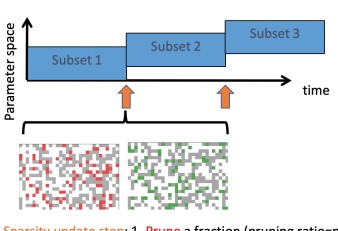

Sparsity update step: 1. Prune a fraction (pruning ratio=pr) of the parameter & 2. re-allocate parameters

Figure 2: Schematic illustration of pruning and re-allocation step in a typical DynSparse training algorithm leading to an evolution of the network representation in parameter space. The dynamic evolution of the sparsity pattern allows the DynSparse training algorithm to explore a larger fraction of the network parameters compared to static sparsity, while remaining "always sparse". For unstructured DynSparse training, the granularity of the sparsity pattern is of block size $1 \times 1$, while for structured DynSparse training, the block size is chosen between $4 \times 4$, $8 \times 8$ and $16 \times 16$.

In the present work, we first study and adapt the unstructured DynSparse training schematically shown in Figure 2 to pre-training of the BERT language models. Specifically, we initialize the sparsity pattern randomly with the same fixed sparsity ratio on all fully connected encoder weights (non-embedding weights). The weights are initialized using a truncated normal distribution (see also Figure 9). During an update step of DynSparse training (see Algorithm 1) we use magnitude pruning to remove a time $t$ dependent fraction $pr(t)$ of the network parameters. The same fraction of parameters is re-allocated elsewhere in the weight tensor. To complete the sparsity update step, all newly allocated parameters and their corresponding first and second-order moments of the Adam optimizer are initialized to zero. Given that DynSparse training has been primarily developed for vision architectures (Dettmers & Zettlemoyer, 2019; Evci et al., 2019) and did not show competitive performance on the language tasks, we find it necessary to reassess some of the algorithm choices for BERT. In particular, during the re-allocation step of DynSparse training, we use random re-allocation of pruned weights instead of gradient-based techniques as in RigL (Evci et al., 2019). For one, this avoids potential issues from a collapse of the explored parameter space (compare Figure 11). More importantly, the absence of dense gradient computation makes our approach always-sparse, such that the full dense model is never actually instantiated. We found that the cosine decay of the pruning

ratio introduced in Evci et al. (2019) outperforms constant pruning schedules and leads to a reduction of the changes in network topology during training. We refer to the maximum pruning ratio $p_r$ simply as "pruning ratio" throughout the paper. All DynSparse hyperparameters are optimized for a sparsity ratio of 0.9 (for more details, refer to Appendix A.1).

## 2.2 BLOCK-SPARSE DYNSPARSE ALGORITHM

Extending unstructured DynSparse training towards using structured sparse computation requires modification to both the prune and update steps in Figure 2. Magnitude pruning can be justified as a simple compression algorithm resulting in unstructured sparsity. However, there is no unique way to extend the magnitude pruning metric to blocks of parameters. Choosing a good metric for block pruning is essential, as magnitude pruning has been surprisingly successful in preserving the task performance of sparse networks (Gale et al., 2019). In the following, we evaluate the selection of norms consisting of $L^\infty$-norm, $L^2$-norm and $L^1$-norm as different criteria for estimating the importance of blocks of parameters. For a block of weights $\mathbf{B} = \mathbf{W}_{\{r,s|(r,s)\in\mathbb{B}\}}$ taken from a weight tensor $\mathbf{W}$ indexed by $\mathbb{B}$, the $L^p$-norm is given by

$$L^p(\mathbf{B}) = \left( \sum_{i,j} |B_{i,j}|^p \right)^{1/p}, \tag{1}$$

where the exponent $p$, e.g. $p = \infty, 2, 1$, controls the relative importance of individual parameters of a block according to their magnitude. In the limit of block size $1 \times 1$, the block pruning according to Eq. (1) reduces to magnitude pruning, allowing us to investigate the task performance with increasing block sizes. For small values of $p \to 0$, each parameter in the block contributes equally towards the importance of the block element as $|B_{i,j}|^p \to 1$, while for large values of $p \to \infty$ the importance of the block collapses towards the parameter with the largest magnitude, with $L^{p\to\infty}(\mathbf{B}) \to \max(|\mathbf{B}|)$. Therefore, the pruning metric for blocks controls the extent to which the magnitude of each of the parameters in a block contributes to the importance of the block itself.

---

**Algorithm 1:** Structured DynSparse training

**Input**: total number of training step $T$, total number of sparsity updates $n - 1$, pruning ratio $p_r(t)$ at time $t$, sparsity ratio $s$, block size $B$;
**Initialize**: Impose random block-sparse pattern on non-embedding weights with uniform constant sparsity ratio across all layers. Initialize weights sampled from a truncated normal distribution;
**for** $k = 1$ **to** $n$ **do**
  Train network with static sparsity pattern for $T/n$ steps;
  For each weight tensor:
  **prune** fraction $p_r(t)$ of blocks with smallest $L^p$-norm defined in Eq. (1);
  **re-allocate** the same fraction $p_r(t)$ of blocks; re-allocated parameters, first and second-order moments of Adam optimizer are all initialized to zero
**end**

---

**Structured regularization**  Group Lasso regularization is commonly used as a structured sparsity-inducing regularizer (Wen et al., 2017; Narang et al., 2017). We introduce the Group Lasso regularization in the update $\Delta\mathbf{W}$ of weight tensor $\mathbf{W}$ following the decoupling of weight decay and Adam optimizer from Loshchilov & Hutter (2017). More specifically, the entry $(i, j)$ of the parameter update $\Delta W_{ij}$ is adjusted to $\Delta W_{ij}^{reg}$ as

$$\Delta W_{ij}^{reg} = \Delta W_{ij} - \mathrm{lr}(t) \cdot \lambda_{group} \cdot w_{std} \cdot \sqrt{B} \frac{W_{ij}}{\sqrt{\sum_{(r,s)\in\mathbb{B}(i,j)} W_{r,s}^2 + \epsilon}}, \tag{2}$$

where $\mathbb{B}(i,j)$ denotes the set of weight indices that belong to the same block as the $(i,j)$th weight element and $\eta(t)$ is the linearly decaying learning rate (see Appendix A). The remaining coefficients are the Group Lasso coefficient $\lambda_{group}$, and the small constant $\epsilon = 10^{-6}$ for numerical stability. The extra pre-factors $w_{std} = 0.02$ and $\sqrt{B}$ corresponding to the standard deviation of the weights at

initialization (see Appendix A) and the square-root of the block size respectively are chosen such as to ensure that the regularization coefficients of weight decay and group lasso are comparable in magnitude.

## 2.3 PARETO CURVE ASSESSMENT OF FLOPs EFFICIENCY

A recent review by Hoefler et al. (2021) pointed out the need for a rigorous framework for comparing sparse training algorithms. In the present work, we introduce a methodology for comparing the sparse task performance on the full BERT-family Pareto curve (Turc et al., 2019), beyond the *Same Capacity Sparse vs. Dense Comparison* approach introduced by Tessera et al. (2021). Comparing different algorithms using a Pareto curve allows to perform a multi-objective assessment under competing constraints, e.g., the desire to use little compute and achieve a high task performance. This multi-objective assessment is particularly useful for assessing the generality and scalability of different training algorithms. Furthermore, the use of Pareto curves allows us to systematically assess algorithmic differences by comparing DynSparse training with dense and static baselines on an equal FLOPs budget.

Choosing optimal learning rates for sparse and dense models of various sparsity ratios and model sizes is essential to ensure a fair comparison of different methods. Naive grid search optimization of the hyperparameters for a full Pareto investigation quickly becomes intractable. To mitigate this, we have identified and tested the applicability of scaling rules for learning rates across model sizes and sparsity ratios.

The dense and sparse BERT-family learning rates are obtained from a grid search for $0.0001 \cdot 2^m$ with $m = 0, 1, ..., 5$ shown in Figure 12 (see Appendix A.4). Interestingly, the results indicate that for a given number of parameters, the optimal learning rate of the sparse model is significantly larger than the learning rates of dense models (Figure 13). To reduce the number of hyperparameter sweeps for large model sizes, we generalize the learning rate scaling with sparsity $s$ as

$$\eta(s) = \eta(s = 0) \exp\left(1.969s^2 + 0.2905s\right), \tag{3}$$

where $\eta(s = 0)$ is the optimal learning rate obtained for a dense model of a given size. We tested the prediction of the unstructured static sparse learning rate fit from Eq. (3) using DynSparse training with block sparsity $16 \times 16$ across both model sizes and sparsity ratio, and obtained good agreement between the predicted optimal sparse learning rate obtained from this rule and values obtained through a grid search, as shown in Figure 13. We also found that the learning rate rule generalizes from static to unstructured DynSparse, as shown in Table A.14. Identifying the mechanism allowing sparse models to profit from larger learning rates than dense models with the same number of parameters (see Figure 13) is left as an exciting direction for future research.

## 3 RESULTS

### 3.1 ADAPTING DYNSPARSE TRAINING TO BERT

In order to establish a general improvement of the dynamic sparse training algorithm with both FLOPs and memory, we study dynamic sparse training of BERT family across multiple model sizes. We analyze the scaling behavior of our DynSparse models with model size (for a fixed sparsity ratio) and sparsity ratio (for a fixed model size).

We find that the DynSparse training algorithm with random re-allocation and sparsity $s = 0.9$ leads to Pareto improvements compared to the dense BERT-family (see Figure 3). The improvements of DynSparse training over the dense baseline remain for a range of model sizes, indicating that DynSparse training can achieve more efficient utilization of FLOPs or network parameters at any scale. Furthermore, we find that these performance advantages are due to the continued updates of the sparsity pattern. We do not observe any improvements of the static baseline in FLOPs efficiency of larger models when the randomly initialized sparsity pattern is kept constant. In fact, for large model sizes, static sparsity almost perfectly matches the dense baseline. This indicates that the sparse network architecture itself brings no performance advantages. Any improvements are therefore expected to arise from continuous compression and evolution of the network representation. For DynSparse BERT-Base, we achieve an improvement in FLOPs efficiency by a factor of 0.48 compared to an interpolation of the dense BERT family, as indicated by the horizontal black arrow in Figure 3.

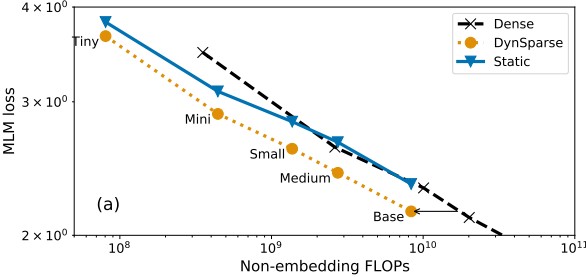 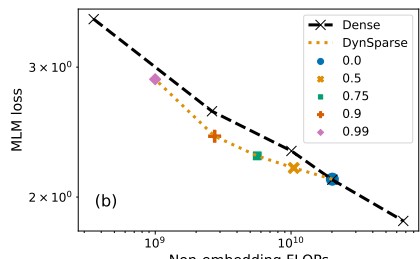

Figure 3: Pareto curve of the BERT family (Turc et al., 2019), comparing validation MLM loss of unstructured DynSparse training (orange dotted line) with static sparsity (solid blue line) and the dense baseline (black dashed line, the standard deviation is not visible at this scale) as a function of FLOPs. All sparsity results are obtained for pre-training with sparsity ratio 0.9, $n = 160$ pattern updates, and optimal pruning ratio $p_r = 0.5$ (see Figure 5). The black arrow indicates a reduction of FLOPs for the same MLM loss by a factor of 0.48.

Figure 4: Comparing validation MLM loss of DynSparse training of BERT-Medium with various sparsity ratios (indicated by color and marker style and joint by orange dotted line) with dense training of BERT family (black dashed line) as a function of non-embedding FLOPs. For all sparsity ratios, we use the hyperparameters optimized for sparsity ratio 0.9.

We observe task performance improvements across a range of sparsity ratios (see Figure 4). However, since the results used hyperparameters tuned for sparsity 0.9, performance for other sparsity ratios could potentially be further improved with additional tuning. In sum, we find that DynSparse training leads to more efficient utilization of parameters and FLOPs for all model sizes.

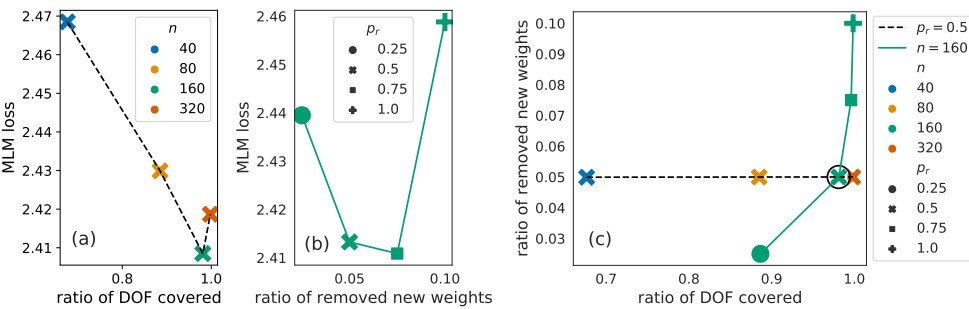

Figure 5: Characterization of the DynSparse pre-training of BERT-Medium with sparsity ratio 0.9. All layer-wise averages shown correspond to maximum value obtained during training. **(a)** MLM loss as a function of the fraction of explored network parameters (DOF) with changing number of sparsity pattern updates $n$. **(b)** MLM loss as a function of the ratio of removed, new weights with changing pruning ratio $p_r$. **(c)** Joint effect of pruning ratio $p_r$ (solid line) on the ratio of removed, new weights, and DOF covered during DynSparse training. The best performing values ($n = 160$, $p_r = 0.5$) from **(a)** are marked by a circle.

To improve our understanding of the sparse training dynamics, we extract measures that can help to explain the efficiency of specific hyperparameter choices (see Appendix A.1). Given that the DynSparse task performance advantage arises from the continual update of the sparsity pattern, we begin by quantifying the amount of parameter exploration. While the DynSparse models have only a tiny fraction of parameters available at any given time, the pattern update means that they can explore all network parameters throughout training and thus increase the effective weight space. We measure the effectively covered space by tracking the fraction of network weights of the corresponding dense network that have been activated at any point during the training and compare with the parameter count of the equivalent dense network to obtain the *total explored degrees of freedom* (DOF)[1]. All

---

[1] A similar quantity has been independently studied in Liu et al. (2021b) as "in-time over-parametrization."

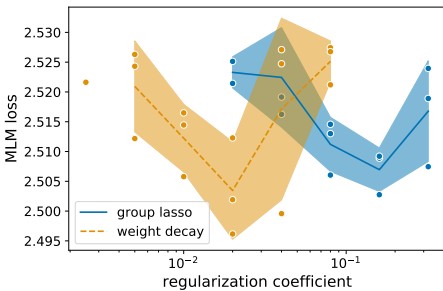
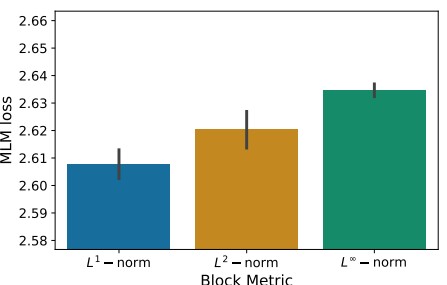

Figure 6: MLM validation loss of DynSparse BERT-Medium with sparsity $s = 0.9$ for block size $B = 16$ as a function of the regularization coefficient for Group Lasso regularization (solid blue) or weight decay (orange dashed). The error bars correspond to the standard deviation over three runs. Number of updates $n = 80$, pruning ratio $p_r = 0.5$.

Figure 7: Block metric dependence of DynSparse training of BERT-Medium with sparsity $s = 0.9$ for block size $B = 16$. Confidence interval is estimated by calculating the standard deviation over three datapoints with their numerical values given in Table A.7.

quantities shown in the following correspond to averages taken over all layers, as we did not observe a systematic layer dependence of these quantities.

The total explored degrees of freedom monotonically increases during training, starting at the fraction of non-zero at the beginning of training and then saturating at an algorithm and hyperparameter dependent value toward the end of training (see Figure 11 for a typical shape). We observe that the maximal number of explored DOF can be controlled through the pruning ratio $p_r$ and the number of sparsity pattern updates $n$ (Figure 5). An increase in the update frequency leads to a simultaneous saturation in both task performance and the number of explored degrees of freedom (Figure 5(a)). On the other hand, the pruning ratio $p_r$ reaches an optimal value and strongly influences the performance with a different fraction of removed, new weights (Figure 5(b)). Notably, we find that the best pruning ratios are reached once the ratio of DOF approaches 1, corresponding to almost complete exploration of all network parameters (Figure 5(c)). Further increases in $p_r$ remove trainable weights that have just been initialized in the previous update step and lead to a deterioration in the task performance. Overall, we note that the best task performance is obtained by balancing the DOF while avoiding wasted compute in the form of parameters that are being allocated and immediately removed (as demonstrated in Figure 5). Given these findings, we postulate that ideal training outcomes require an *exploration of all available parameters* as well as an only *moderate amount of noise injection*.

## 3.2 BLOCK-SPARSE DYNSPARSE TRAINING

In structured DynSparse training, the block pruning is done according to the $L^p$-norms from Eq. (1) of the respective blocks of parameters. For the $L^p$-norms norms studied ($p = 1, 2, \infty$), as shown in Table 7 we obtain the best performance using the $L^1$-norm, which corresponds to the sum of the parameter magnitudes. Moreover, all block parameters contribute toward the block's importance, given that $L^1$-norm outperforms other norms that assign larger importance to dominating weights in a block.

Next, we evaluate the use of structured regularization applied to sparse weights during DynSparse training with block size 16×16. To compare potential advantages from using a structured regularization against an unstructured regularization method, we have also evaluated the task performance for tuning weight decay coefficient instead of the Group Lasso coefficients. As shown in Figure 6, we obtain the best task performance using weight decay. The regularization coefficients are only tuned for the sparse, non-embedding weights. Other sources of unstructured regularization such as dropout (and in the case of Group Lasso also weight decay) are set to zero. While our results are in agreement with the competitiveness of block pruning versus the Group Lasso experiments in Narang et al. (2017), we have not tested more advanced regularization methods (Yang et al., 2019; Mummadi et al., 2019). We find that the structured regularization does not lead to any performance advantages over tuning weight decay.

Table 1: Task performance of DynSparse training of BERT-Base with sparsity $s = 0.9$ for various block sizes $B$, compared to dense BERT-Small with similar number of FLOPs and linear interpolation of baseline values ("Matched") with exactly the same number of FLOPs. Hyperparameters are not specifically tuned for $B = 16$ (number of updates $n = 80$, pruning ratio $p_r = 0.5$). See Appendix Table A.8 for block size dependence. The standard deviation is estimated over three runs.

| Model | $B$ | MLM | FLOPs |
|---|---|---|---|
| Small (dense) | - | $2.310 \pm 0.002$ | $10.07 \cdot 10^9$ |
| Matched (dense) | - | $2.350$ | $8.33 \cdot 10^9$ |
| Base ($s = 0.9$) | 16 | $2.311 \pm 0.01$ | $8.33 \cdot 10^9$ |
| Base ($s = 0.9$) | 8 | $2.295 \pm 0.002$ | $8.33 \cdot 10^9$ |
| Base ($s = 0.9$) | 4 | $2.272 \pm 0.01$ | $8.33 \cdot 10^9$ |
| Base ($s = 0.9$) | 1 | $2.160 \pm 0.002$ | $8.33 \cdot 10^9$ |

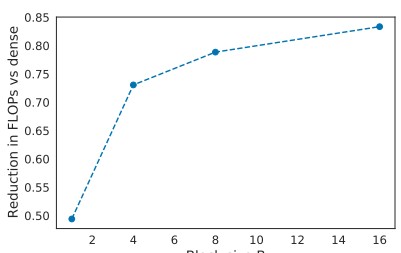

Figure 8: Block size dependence of the reduction in FLOPs for DynSparse training compared to (interpolation) of the dense BERT family for a given task performance. Values correspond to the block sparse DynSparse training of BERT-Base given in Table 1.

Next, we compare structured DynSparse training against the baseline using both horizontal and vertical slice of a Pareto plot. For a vertical slice, e.g., a constant FLOPs comparison, we demonstrate that DynSparse training can preserve some task performance advantages when block sparsity of size 4×4, 8×8, and 16×16 is used (Table 1). For a horizontal slice (see horizontal arrow in Figure 3) measuring the reduction in FLOPs for a given task performance, we achieve a reduction between 0.5 for unstructured sparsity and 0.83 for block size $B = 16$, as shown in Figure 8. The inverse of the FLOPs efficiency improvements gives the maximum relative execution time of FLOPs for sparse compared to dense computation to preserve Pareto improvements in terms of wallclock time and needs to be below a factor of 2 for unstructured sparsity and 1.2 for block sparsity for DynSparse training (see Appendix A.3). This compute efficiency makes DynSparse training promising for practical applications that seek to further benefit from the higher computational efficiency of block computation.

## 4 RELATED WORK

**Lottery ticket and pruning at initialization**   Weight sparsity has been traditionally viewed as a technique for compressing network representation leading to reduced FLOPs and trainable parameters. The lottery ticket hypothesis (Frankle & Carbin, 2019; Frankle et al., 2020) postulates that through iteratively pruning and re-initialization, it is often possible to identify smaller subnetworks at initialization or early on during training that can be trained to full model performance of a large over-parametrized network. Since then, there has been a significant amount of work studying techniques for identifying sparsity distributions at initialization (Lee et al., 2019; Wang et al., 2020; Tanaka et al., 2020; Zhang & Stadie, 2020; Lee et al., 2020; Frankle et al., 2021; Su et al., 2020). Recently, the identification of lottery tickets early during training has allowed achieving time-to-train savings after a short, dense training phase (Chen et al., 2021) by pruning attention heads and neurons early during the pre-training phase (Chen et al., 2021).

**Dynamic sparsity**   In DynSparse (Mocanu et al., 2018; Bellec et al., 2017; Liu et al., 2019; Mostafa & Wang, 2019; Dettmers & Zettlemoyer, 2019; Evci et al., 2019; Liu et al., 2021a), the sparse connectivity pattern is evolved during training. Most DynSparse algorithms currently rely on magnitude pruning to remove unimportant network parameters. However, the algorithm show large differences in the exact re-allocation criteria, which range from random re-allocation (Bellec et al., 2017; Mocanu et al., 2018; Liu et al., 2019; 2021a) to a directed evolution based on momentum (Dettmers & Zettlemoyer, 2019) or gradients (Evci et al., 2019).

**Compute-efficient sparse training**   Complementary to viewing weight sparsity as a compression technique of dense networks, sparsity allows increasing network dimensions, potentially resulting in an augmentation of the effective model capacity for a given amount of compute and memory (Gray et al., 2017). However, most investigations into sparse training currently impose algorithmic con-

straints through the use of pre-defined sparsity patterns (Vooturi et al., 2020; Zhou et al., 2021), coarse-grained sparsity structures (Gray et al., 2017) or even result in increased compute and memory compared to dense training through the use of masking.

In the present work, we contribute towards compute-efficient training from an algorithmic point of view, by extending DynSparse training towards structure. Additionally, we leverage the 2nd generation of Graphcore's Intelligence Processing Unit (IPU) (Graphcore, 2021) to dynamically train large, structured DynSparse models using Graphcore's DynSparse library[2].

**Structured sparsity**   Simple unstructured sparse training algorithms based on magnitude pruning heuristic have shown remarkable ability to preserve the task performance of over-parametrized neural networks (Gale et al., 2019). Nevertheless, on the execution side, unconstrained magnitude pruning results in unstructured sparsity patterns, which remain challenging to support on traditional hardware accelerators (Narang et al., 2017). Using coarser-grained sparsity structures resulting in contiguous memory access can mitigate this problem. Nevertheless, the resulting gains in execution efficiency are often achieved at the cost of a deterioration in task performance (Narang et al., 2017; Mostafa & Wang, 2019). Approaches to improve the task performance of structured sparsity during training range from structured regularization (Wen et al., 2017; Narang et al., 2017; Yang et al., 2019; Mummadi et al., 2019; Louizos et al., 2018), threshold-based pruning using representation based on block sparsity (Narang et al., 2017), network slimming (Liu et al., 2017) and low-rank factorization (Wang et al., 2019) to frequently changing sparsity pattern with distinct granularity varying from block (Hadifar et al., 2020), channel (Gao et al., 2018) to full layers (Fan et al., 2020).

**Conditional sparsity and models with large number of parameters**   Unlike dynamic sparsity, conditional sparsity does not reduce the number of trainable parameters that define the model. The task performance of semi-supervised language models generally improves with model size under appropriate scaling of total computation time and dataset size (Kaplan et al., 2020). In conditional sparse training (Shazeer et al., 2017; Lepikhin et al., 2020; Fedus et al., 2021; Lewis et al., 2021), activations are dynamically routed to subsets of the network weights distributed over a large number of hardware accelerators. Conditional sparse training leverages increases in the number of network parameters to improve the task performance for a constant FLOPs budget (Fedus et al., 2021).

## 5   CONCLUSION & FUTURE WORK

In this work, we demonstrated that DynSparse training of BERT leads to a more FLOP-efficient utilization of the trainable parameters. Our experimental work has focused on BERT MLM pre-training with sequence length 128, and further research is needed to evaluate the performance of pre-training with larger sequence lengths and fine-tuning to downstream tasks.

An important direction stems from the practical opportunity to translate the FLOPs savings into reduced cost of training. Remarkably, we found that even a naive block-sparse version of the DynSparse algorithm remains FLOP-Pareto efficient, which forms the first step towards more compute-efficient training of large-scale language models. However, further task performance improvements are necessary to fully translate the task performance advantages into time-to-train win on the Pareto curve. In particular, it will be important to shed further light on the conditions that enable the performance gains in unsupervised training, particularly the relationship between the number of available parameters and achievable task performance.

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

## A TECHNICAL DETAILS

- **Optimizer:** Throughout this work we use element-wise optimization based on Adam with weight decay 0.01, $\beta_1 = 0.9$, $\beta_2 = 0.999$, $\epsilon = 10^{-6} \times$ loss-scaling-factor and gradient clipping, which is known to work well with the sparse gradients found in NLP models.

- **Default learning rate schedule** consists of 10000 linear warmup steps up to the maximum learning rate (0.0002 for BERT-Medium and 0.0001 for BERT-Base), followed by a linear decay over the full training run.

- **Default dropout** is 0.1 for all models larger then BERT-Small. To avoid artificial performance gains through an adjustment of the regularizer in the presence of sparsity induced regularization (Bartoldson et al., 2019), we keep dropout in the sparse models identical to the one used in the corresponding baseline.

- **Default floating-point precision**: We use datatype FP16.16 (16 bit compute with 16 bit partials) throughout the model. The second-order moment in the Adam optimizer is computed and stored in FP32. Embedding is kept in FP16. The default loss-scaling factor for both BERT-Medium and BERT-Base is 512.

- **Initialization scheme**: The sparsity pattern is initialized randomly. The weights are initialized using a truncated normal initializer with an initialization range of $w_{std} = 0.02$. This choice was motivated by having compared different initializations for the sparse model and found that the dense default truncated normal gives the best task performance, as shown in Figure 9. We found that preserving the variance of the activation statistics of the sparse model compared to the dense model (Evci et al., 2020) does not lead to any performance gains.

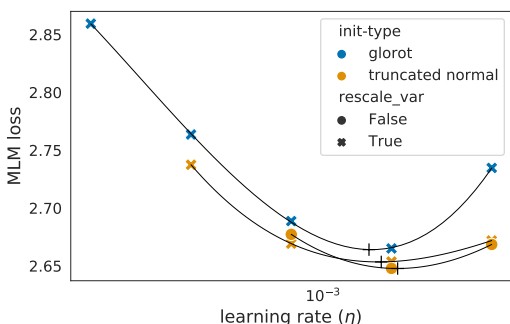

Figure 9: BERT-Medium with static unstructured sparsity $s = 0.9$ imposed on all weights using Glorot (blue) or truncated normal (orange) initialization scheme. The marker shape indicates whether the standard deviation of the weight initializiation was increased.

- **Pre-training dataset**: Phase I pre-training is performed on Wikipedia and BookCorpus using Whole Word Masking with a sequence length of 128.

- Code for DynSparse library used in this work is available at Graphcore's Github `https://github.com/graphcore/examples/tree/master/applications/tensorflow/dynamic_sparsity`

### A.1 HYPERPARAMETERS SPECIFIC TO DYNAMIC SPARSITY (DYNSPARSE)

Two very important hyper-parameters for DynSparse training are the sparsity pattern update frequency, i.e. how often the network topology is modified, and the pruning ratio, which determines the fraction of the network topology modified at each update step. The sparsity ratio per layer is kept fixed throughout training.

- **Update frequency dependence**: Comparing the task performance of 20, 40, 80, 160 and 320 updates at sparsity ratio 0.9, we have found that the task performance improves with the number of sparsity pattern updates (Table A.2 and A.4). We chose the optimal number of updates as $n = 160$ ($n = 80$) for sparsity ratio 0.9 and block size 1×1 (16×16).

- **Extreme sparsity regime**: All hyperparameters have been optimized for sparsity 0.9. However, we tested how well the pruning ratio and update frequency dependence optimized

Table A.2: Number of sparsity pattern updates $n$ dependence of unstructured ($1\times1$) DynSparse BERT-Medium, $\eta = 0.001397$, sparsity $s = 0.9$, 10 epochs, phase I (pruning ratio $p_r = 0.5$ with cosine decay and random reallocation).

| $n$ | MLM loss | NSP |
|-----|----------|------|
| 40 | 2.468 | 0.645 |
| 80 | 2.430 | 0.656 |
| 160 | **2.409** | 0.626 |
| 320 | 2.419 | 0.649 |

Table A.3: Pruning ratio $p_r$ dependence of unstructured ($1\times1$) DynSparse BERT-Medium, $\eta = 0.001397$, sparsity $s = 0.9$, 10 epochs, phase I (number of updates $n = 160$ with cosine decay and random reallocation). Same hyperparameters as in Table A.2.

| $p_r$ | MLM loss | NSP loss |
|-------|----------|----------|
| 0.25 | 2.439 | 0.655 |
| 0.50 | 2.413 | 0.684 |
| 0.75 | 2.411 | 0.668 |
| 1.00 | 2.459 | 0.698 |

Table A.4: Number of sparsity pattern updates $n$ dependence of structured ($16\times16$) DynSparse BERT-Medium, $\eta = 0.001397$, sparsity $s = 0.9$, 10 epochs, phase I (pruning ratio $p_r = 0.5$ with cosine decay and random reallocation).

| $n$ | MLM loss | NSP loss |
|-----|----------|----------|
| 40 | 2.616 | 0.731 |
| 80 | 2.606 | 0.650 |
| 160 | 2.633 | 0.692 |
| 320 | 2.645 | 0.693 |

Table A.5: Pruning ratio $p_r$ dependence of structured ($16\times16$) DynSparse BERT-Medium, $\eta = 0.001397$, sparsity $s = 0.9$, 10 epochs, phase I (number of updates $n = 160$ with cosine decay and random reallocation). Same hyperparameters as in Table A.4.

| $p_r$ | MLM loss | NSP loss |
|-------|----------|----------|
| 0.25 | 2.648 | 0.694 |
| 0.50 | **2.633** | 0.692 |
| 0.75 | 2.634 | 0.745 |
| 1.00 | 2.675 | 0.701 |

Table A.6: Pruning ratio $p_r$ and number of update $n$ dependence of unstructured ($1\times1$) DynSparse BERT-Medium, $\eta = 0.001837$, sparsity $s = 0.99$, 10 epochs, phase I (with cosine decay and random reallocation).

| $s$ | $n$ | $p_r$ | MLM loss | NSP loss |
|------|-----|-------|----------|----------|
| 0.99 | 160 | 0.10 | 2.999 | 0.833 |
| 0.99 | 160 | 0.25 | 2.939 | 0.789 |
| 0.99 | 160 | 0.50 | 2.889 | 0.750 |
| 0.99 | 160 | 0.75 | **2.872** | 0.775 |
| 0.99 | 80 | 0.50 | 2.922 | 0.842 |
| 0.99 | 160 | 0.50 | 2.889 | 0.750 |
| 0.99 | 320 | 0.50 | 2.868 | 0.772 |
| 0.99 | 640 | 0.50 | 2.886 | 0.791 |

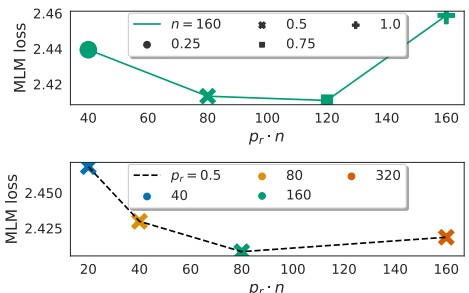

Figure 10: MLM loss vs pruning ratio $p_r$ times number of sparsity pattern updates $n$ for unstructured DynSparse training of BERT-Medium with sparsity ratio 0.9 for different values of (**Top panel**) pruning ratio $p_r$ (with $n = 160$) and (**Bottom panel**) sparsity pattern updates $n$ (with $p_r = 0.5$). Same data as in Figure 5.

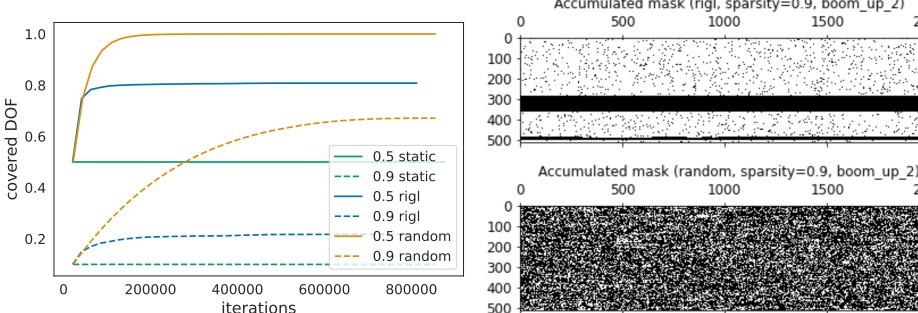

Figure 11: (**Left panel**) Fraction of explored degrees of freedom for static sparsity and unstructured DynSparse training using gradient based (RigL) (Evci et al., 2019) vs random re-allocation (Dettmers & Zettlemoyer, 2019). (**Right panel**) Corresponding sparsity patterns for the first up-projection in the feedfoward component ("Boom-up") of the second transformer block, accumulated throughout training, for sparsity ratio 0.9 using gradient based (RigL) and random based reallocation. A black (white) dot corresponds to a parameter being non-zero (zero) at any point during training. The dark horizontal blocks in the RigL updates indicate a collapse due to outliers along the input dimension, which indicates that the effect arises from the activation part of the dense gradient update. This suggests that the collapse could be mitigated by reducing the influence of the activations during DynSparse training update.

for sparsity 0.9 translates to sparsity 0.99. We have found that increasing the pruning ratio to $p_r = 0.75$ can lead to small performance gains, as shown in Table A.6.

- **Total number of pruned parameters** The pruning ratio $p_r$ and the number of updates $n$ jointly control the total number of pruned and re-allocated parameters. The total number of pruned and re-allocated parameters is proportional to their product. We obtain an optimal value of their product in terms of task performance as shown in Figure 10.

- **Re-allocation criteria**: We found that random re-allocation outperforms gradient-based re-allocation. While the pruning criterion leads to a compression of the network topology, the growing criterion directs the evolution of the network topology and distinguishes DynSparse training as a form of neural architecture search during training from mere gradual pruning approaches. Understanding the requirements for efficient joint subspace exploration of parameter and network topology space using DynSparse training will be essential to scale towards larger language models. In Figure 11, we show that for gradient-based re-allocation, the dense gradient is dominated by outliers in the activation, e.g., along the input dimension of each layer, which imposes a strong bias on the available degrees of freedom during the update step. In agreement with this observation, we find that for random-based re-allocation, a significantly larger fraction of the network parameters is explored during training, while for gradient-based re-allocation the training remains constrained into a small subset of all network parameters (left panel of Figure 11).

- **Block size metric** The importance of blocks of parameters is assessed by evaluating the $L^1$-norm of the corresponding blocks (see Table A.7 and Figure 7).

Table A.7: Task performance of DynSparse training of BERT-Medium with sparsity 0.9 for block size $B = 16$ for various block size metrics.

| Block metric | MLM loss | NSP loss |
|---|---|---|
| $L^2$-norm | 2.611 | 0.684 |
| $L^2$-norm | 2.623 | 0.684 |
| $L^2$-norm | 2.627 | 0.664 |
| $L^\infty$-norm | 2.632 | 0.686 |
| $L^\infty$-norm | 2.635 | 0.720 |
| $L^\infty$-norm | 2.637 | 0.670 |
| $L^1$-norm | 2.603 | 0.665 |
| $L^1$-norm | 2.606 | 0.650 |
| $L^1$-norm | 2.615 | 0.731 |

Table A.8: Task performance of DynSparse BERT-Medium with sparsity 0.9 for various block sizes $B$ compared to dense BERT-Mini with similar number of FLOPs and linear interpolation of the baseline values ("Matched") with exaclty the same number of FLOPs. Hyperparameters are not specifically tuned for different block sizes. See also BERT-Base results in Table 1.

| Model | $B$ | MLM | FLOPs |
|---|---|---|---|
| Mini | - | 2.614 | $2.617 \cdot 10^9$ |
| Matched (dense) | - | 2.603 | $2.738 \cdot 10^9$ |
| Medium ($s = 0.9$) | 16 | 2.621 | $2.738 \cdot 10^9$ |
| Medium ($s = 0.9$) | 8 | 2.591 | $2.738 \cdot 10^9$ |
| Medium ($s = 0.9$) | 4 | 2.546 | $2.738 \cdot 10^9$ |
| Medium ($s = 0.9$) | 1 | **2.408** | $2.738 \cdot 10^9$ |

Table A.9: MLM validation loss of BERT-Small for results given in Figure 1.

| s | type | $\eta$ | MLM loss | NSP loss |
|---|---|---|---|---|
| 0.25 | zero | 0.000343 | 2.390 | 0.653 |
| 0.50 | zero | 0.000589 | 2.485 | 0.687 |
| 0.75 | zero | 0.001011 | 2.637 | 0.737 |
| 0.90 | zero | 0.001397 | 2.829 | 0.802 |
| 0.99 | zero | 0.001697 | 3.244 | 0.907 |
| 0.25 | untrained | 0.000686 | 2.375 | 0.681 |
| 0.50 | untrained | 0.001178 | 2.491 | 0.675 |
| 0.75 | untrained | 0.002021 | 2.645 | 0.731 |
| 0.90 | untrained | 0.002795 | 2.850 | 0.829 |
| 0.99 | untrained | 0.003394 | 3.243 | 0.827 |

- **Block size dependence** The block size dependence of BERT-Medium with sparsity 0.9 is given in Table A.8.
- **Untrainable vs zero-valued parameters** Numerical values for the results shown in the left panel of Figure 1 are given in Table (A.9).

## A.2  SPARSE FLOPS: FLOPS ESTIMATES FOR SPARSE MULTIPLICATION WITH DENSE INPUT

Throughout this report we assume the FLOPs for training a dense layer with sparse weight elements to approximately scale as $\mathcal{O}(3 \times 2 \times I \times B \times O \times f)$, where $B$ the batch dimension, $I$ is the input dimension, $O$ the output dimension and $f$ is the density of sparsity pattern imposed on the corresponding dense layer, which is to the sparsity ratio $s$ as $f = 1 - s$. The FLOPs estimate can be divided into the following components:

1. **FLOPs estimate for sparse forward pass:** Assuming a sparse matrix $\mathbf{M}$ has a sparsity ratio $s$ or a density $f = 1 - s$, the required matrix multiplication for a given dense input $\vec{x}$ and output $\vec{y}$ is

$$y_{bi} = \sum_{j|M_{ij} \neq 0} M_{ij} x_{bj}, \qquad (4)$$

where $\mathbf{M}$ has dimension $[O, I]$ and $\dim(y) = [B, O]$, $\dim(x) = [B, I]$,

   (a) **Sparse Multiplication**: performing the summation $z_{bij} = M_{ij} x_{bj}$ for $i, j$ **iff** $M_{ij} \neq 0$ gives us a reduction of the total number of FLOPs by a fraction of non-zero elements in $\mathbf{M}$ times $B$ leading to $B \times O \times I \times f$ FLOPs.

(b) **Sparse Addition**: performing $\sum_j z_{bij}$ requires us to calculate the exact number of non-zeros along the input dimension, giving $B \times O \times \text{prob}(out) \times I \times \text{prob}(in) - B \times O \times \text{prob}(out)$, where we defined some probability for non-zero values along the output dimension $\text{prob}(out)$ and input dimension $\text{prob}(in)$. Assuming a uniform distribution, we estimate the FLOPs count to scale approximately linearly with the sparsity ratio $B \times O \times I \times f - B \times O \times f/\text{prob}(in)$ to first order.

The total FLOPs of sparse multiplication used in the forward pass scales approximately linearly in the number of non-zeros, i.e. $\mathcal{O}(2I \times B \times O \times f)$.

2. **FLOPs estimate for recursive propagation of error through the network**: Involves a multiplication of the dense error with the transposed sparse matrix leading $\mathcal{O}(2I \times B \times O \times f)$ additional FLOPs.

3. **FLOPs estimates for the outer product** The weight update itself is formed by a sparse outer product, where only the sparse components need to be updated, which leads to a further reduction in the number of FLOPs that scales linearly with the density of the matrix.

## A.3 RELATING IMPROVEMENTS IN FLOPS EFFICIENCY TO IMPLEMENTATION REQUIREMENTS

To relate the algorithmic improvement in FLOPs efficiency to implementation requirements in general hardware agnostic way, we consider the sparse extra cost $\varepsilon$ that we define as

$$\varepsilon := \frac{\Delta t^{\text{sparse}}}{\Delta t^{\text{dense}}}, \tag{5}$$

where $\Delta t^i$ ($i = \text{sparse, dense}$) is the average time it takes to execute a FLOP of type i for a specific model size and sparsity ratio. For a given fixed number of training steps and the **same** task performance, DynSparse training with theoretical FLOPs $F^{\text{sparse}}$ (defined in Appendix A.2) is only faster than dense training with FLOPs $F^{\text{dense}}$ if the time to execute a sparse training step $t^{sparse}$ is smaller than the time to execute a dense training step $t^{dense}$. Or formally:

$$t^{sparse} < t^{dense}, \qquad F^{\text{sparse}} \Delta t^{\text{sparse}} \quad < F^{\text{dense}} \Delta t^{\text{dense}}. \tag{6}$$

In other words, the utilization of fewer but "slower" FLOPs in the sparse model still translates to a faster execution of the sparse model overall. Note that this comparison is performed at equal task performance and for the same number of training steps.

Using this formalism, we can view improvements in task performance in the context of throughput requirements for a given algorithm in a hardware agnostic way independent of the exact implementation. For $t^{sparse} = t^{dense}$, we can derive the maximum critical extra cost for a DynSparse implementation before DynSparse training loses its advantages over dense computation in terms of time-to-train win. Specifically, for a given fixed number of training step and **same** task performance, the critical cost factor is given by

$$\varepsilon_{critical} = F^{\text{dense}} / F^{\text{sparse}} \tag{7}$$

where $F^{\text{sparse}}$ corresponds to the DynSparse training and $F^{\text{dense}}$ to the (interpolated) dense BERT family. We emphasize, that besides this requirement for a time-to-train win sparse training also allows to run models with larger model dimensions for a given number of parameters.

## A.4 LEARNING RATE FOR SPARSE AND DENSE MODELS

The results of the learning rate sweep of BERT with various sparsities are given in Table A.10. The corresponding learning rate sweep for the dense BERT-family is given in Table A.11. We confirmed that the optimal learning rates for static sparsity agree with the ones for DynSparse in Table A.14. We confirmed that the predicted learning rate dependence of the DynSparse model generalizes to block sparsity and across multiple model sizes as given in Table A.12 for block sparsity 16x16.

### A.4.1 LEARNING RATE FOR SPARSE MODELS

In Figure 12, we show the learning rate sweep of the BERT-Medium model with static sparsity, for various sparsity ratios. We estimate the optimal learning rate for sparse models through the minimum of a cubic interpolation of the task performance vs learning rates for a given sparsity ratio, as indicated

Table A.10: Learning rate ($\eta$) sweep of static unstructured sparsity BERT-Medium, sparsity $s = 0, 0.25, 0.5, 0.75, 0.9$.

| $\eta$ | sparsity | MLM loss | NSP loss |
|---|---|---|---|
| 0.0001 | 0.00 | 2.179 | 0.610 |
| 0.0002 | 0.00 | 2.115 | 0.598 |
| 0.0002 | 0.00 | 2.115 | 0.605 |
| 0.0004 | 0.00 | 2.116 | 0.606 |
| 0.0008 | 0.00 | 2.164 | 0.633 |
| 0.0001 | 0.25 | 2.278 | 0.627 |
| 0.0002 | 0.25 | 2.204 | 0.642 |
| 0.0004 | 0.25 | 2.186 | 0.596 |
| 0.0008 | 0.25 | 2.223 | 0.638 |
| 0.0001 | 0.50 | 2.412 | 0.679 |
| 0.0002 | 0.50 | 2.338 | 0.671 |
| 0.0004 | 0.50 | 2.283 | 0.631 |
| 0.0008 | 0.50 | 2.298 | 0.648 |
| 0.0002 | 0.75 | 2.551 | 0.741 |
| 0.0004 | 0.75 | 2.483 | 0.685 |
| 0.0008 | 0.75 | 2.446 | 0.671 |
| 0.0016 | 0.75 | 2.449 | 0.647 |
| 0.0032 | 0.75 | 2.547 | 0.707 |
| 0.0004 | 0.90 | 2.723 | 0.758 |
| 0.0008 | 0.90 | 2.677 | 0.711 |
| 0.0016 | 0.90 | 2.648 | 0.706 |
| 0.0032 | 0.90 | 2.669 | 0.697 |

Table A.11: Learning rate ($\eta$) sweep for dense BERT-family consisting of BERT-Tiny, Mini, Small, Medium and Base.

| model | $\eta$ | MLM loss | NSP loss |
|---|---|---|---|
| Mini | 0.000050 | 3.062 | 0.839 |
| Mini | 0.000100 | 2.833 | 0.811 |
| Mini | 0.000400 | 2.625 | 0.742 |
| Mini | 0.000800 | 2.606 | 0.775 |
| Mini | 0.001600 | 2.628 | 0.779 |
| Mini | 0.003200 | 2.665 | 0.783 |
| Small | 0.000800 | 2.326 | 0.644 |
| Small | 0.000400 | 2.310 | 0.621 |
| Small | 0.000200 | 2.329 | 0.635 |
| Small | 0.001600 | 2.418 | 0.768 |
| Medium | 0.000200 | 2.115 | 0.605 |
| Medium | 0.000400 | 2.116 | 0.606 |
| Medium | 0.000800 | 2.164 | 0.633 |
| Medium | 0.000100 | 2.179 | 0.610 |
| Base | 0.000025 | 2.115 | 0.599 |
| Base | 0.000100 | 1.878 | 0.542 |
| Base | 0.000050 | 1.972 | 0.569 |
| Base | 0.000200 | 1.843 | 0.488 |

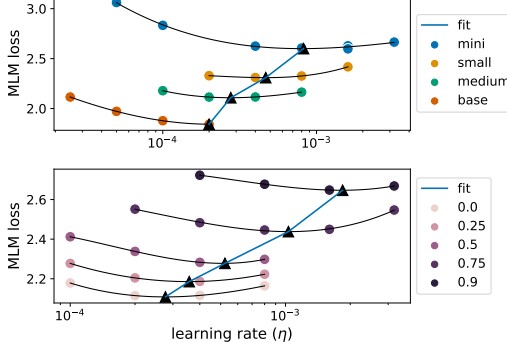

Figure 12: MLM validation loss for the **Top panel** dense BERT family **Bottom panel** static BERT of different sparsity ratios between 0 and 0.9 as a function of learning rate. The solid line correspond to a cubic fit for all data with the same sparsity ratio. The minimum of the resulting fit corresponds to the optimal learning rate for a given sparsity and is indicated by the black triangles connected by blue lines.

Table A.12: Learning rate ($\eta$) sweep for DynSparse BERT-family BERT-Mini, Small, Medium and Base for sparsity 0.9 with block size 16×16.

| model | $\eta$ | MLM | NSP |
|---|---|---|---|
| Base | 0.000160 | 2.520 | 0.692 |
| Base | 0.000320 | 2.429 | 0.693 |
| Base | 0.000640 | 2.340 | 0.647 |
| Base | 0.001280 | 2.328 | 0.603 |
| Base | 0.002560 | 2.369 | 0.656 |
| Medium | 0.000125 | 2.878 | 0.892 |
| Medium | 0.000250 | 2.760 | 0.720 |
| Medium | 0.000500 | 2.670 | 0.730 |
| Medium | 0.002000 | 2.640 | 0.715 |
| Mini | 0.001250 | 3.184 | 0.882 |
| Mini | 0.002500 | 3.145 | 0.871 |
| Mini | 0.005000 | 3.147 | 0.869 |
| Mini | 0.005120 | 3.195 | 0.907 |
| Small | 0.000313 | 2.927 | 0.865 |
| Small | 0.000625 | 2.841 | 0.773 |
| Small | 0.001250 | 2.788 | 0.861 |
| Small | 0.005000 | 2.826 | 0.797 |

Table A.13: Learning rate sweep of BERT-Small alternating between dense and sparse training with either non-trainable parameter or zero-valued parameter corresponding to sparsity $s = 0.9$, 10 epochs phase I, for various pruning methods. Optimal values are given in Table A.16. We switch $n = 160$ times between sparse/non-trainable parameters and the dense training.

| non active | pruning | $\eta$ | MLM | NSP |
|---|---|---|---|---|
| non-train | fixed | 0.0002 | 2.366 | 0.671 |
| non-train | fixed | 0.0004 | **2.358** | 0.668 |
| non-train | fixed | 0.0008 | 7.242 | 0.693 |
| non-train | magnitude | 0.0002 | 2.379 | 0.658 |
| non-train | magnitude | 0.0004 | **2.354** | 0.675 |
| non-train | magnitude | 0.0008 | 11.160 | 0.766 |
| non-train | random | 0.0001 | 2.431 | 0.733 |
| non-train | random | 0.0002 | 2.365 | 0.669 |
| non-train | random | 0.0004 | **2.349** | 0.693 |
| non-train | random | 0.0008 | 7.272 | 0.693 |
| zero | fixed | 2.5e-05 | 3.317 | 0.967 |
| zero | fixed | 5e-05 | **3.199** | 0.817 |
| zero | fixed | 0.0001 | 3.277 | 0.819 |
| zero | fixed | 0.0002 | 3.329 | 0.884 |
| zero | fixed | 0.0004 | 3.358 | 0.964 |
| zero | fixed | 0.0008 | 3.424 | 0.799 |
| zero | magnitude | 0.0002 | 2.746 | 0.756 |
| zero | magnitude | 0.0004 | **2.685** | 0.711 |
| zero | magnitude | 0.0008 | 3.056 | 0.834 |
| zero | magnitude | 0.0016 | 6.538 | 1.217 |
| zero | random | 0.0001 | 6.232 | 1.142 |
| zero | random | 0.0002 | 6.132 | 1.273 |
| zero | random | 0.0004 | **6.094** | 1.185 |
| zero | random | 0.0008 | 6.284 | 0.987 |

Table A.14: Learning rate sweep of DynSparse BERT-Medium, sparsity $s = 0.9$, 10 epochs phase I, used to confirm that the optimal learning rates for static sparsity from Table A.10 translate into optimal learning rates for DynSparse.

| $\eta$ | MLM loss | NSP loss |
|---|---|---|
| 0.00064 | 2.467 | 0.647 |
| 0.00128 | 2.410 | 0.670 |
| 0.0026 | 2.429 | 0.674 |
| 0.0051 | 2.521 | 0.654 |

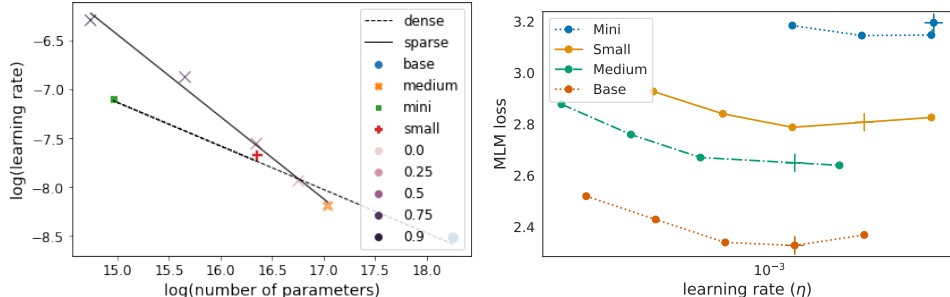

Figure 13: (**Left panel**) Dense fit to the optimal learning rate estimated as the position of the black triangles from Figure 12 for BERT-Medium with various sparsities and the dense BERT-family as a function of the number of trainable parameters $N$ for various model sizes (indicated by symbol style and color) and sparsity ratios (colored crosses). The black lines indicate linear fits that are best approximated by $\log(\eta) = -0.8383(\pm 0.05)\log(N) + 6.13(\pm 0.7)$ for the sparse models and $\log(\eta) = -0.44(\pm 0.05)\log(N) - 0.47(\pm 0.9)$ for the dense models. (**Right panel**) Testing the prediction of the optimal sparse learning rate from Eq. 3 (markerstyle "+") on BERT-family with sparsity 0.9 and block size $16 \times 16$ (values given in Table A.12).

Table A.15: Learning rate sweep of DynSparse BERT-Small unfreeze (freeze) experiment with initial (final) fraction of non-trainable parameters 0.9, 10 epochs phase I.

| type | $\eta$ | MLM loss | NSP loss |
|------|--------|----------|----------|
| freeze | 0.000087 | 2.467 | 0.715 |
| freeze | 0.000175 | **2.407** | 0.703 |
| freeze | 0.000349 | 2.420 | 0.685 |
| freeze | 0.000699 | 2.540 | 0.695 |
| unfreeze | 0.000175 | 2.933 | 0.666 |
| unfreeze | 0.000349 | 2.598 | 0.676 |
| unfreeze | 0.000699 | **2.440** | 0.703 |
| unfreeze | 0.001397 | 7.251 | 0.693 |
| unfreeze | 0.002795 | 7.520 | 0.784 |

Table A.16: MLM validation loss of BERT-Small trained by alternating between dense and training only a fraction of 0.1 of the non-embedding weights with the non-trainable parameter set to either zero or just untrainable without modifications. We pick the best learning rate for each experiment using a grid search over $2.5 \cdot 10^{0.5} \cdot 2^m$ with $m = 0, 1, 2, ...$ (Table A.13) (number of updates $n = 160$).

| $\eta$ | non-train | selection | MLM |
|--------|-----------|-----------|-----|
| 5e-05 | zero | fixed | 3.199 |
| 0.0004 | zero | magnitude | **2.685** |
| 0.0004 | zero | random | 6.094 |
| 0.0004 | untrained | fixed | 2.358 |
| 0.0004 | untrained | magnitude | 2.354 |
| 0.0004 | untrained | random | **2.349** |

by the triangle markers in Figure 12. We find that the optimal learning rate $\eta$ calculated from the interpolation is best approximated by

$$\log(\eta(s)) \approx 1.969(\pm 0.2)s^2 + 0.2905(\pm 0.2)s - 8.175(\pm 0.04) \tag{8}$$

as a function of sparsity $s$ or equivalently as (see Figure 12)

$$\log(\eta(N)) \approx -0.838(\pm 0.05)\log(N) + 6.13(\pm 0.73) \tag{9}$$

for number of parameters $N$. Interestingly, a linear learning rate vs logarithmic memory fit as used in Kaplan et al. (2020) ($\eta(N) \approx 0.003239 - 0.0001395\log(N)$ from Eq. (D1)) is leading to qualitatively worse agreement, which might be explained by our optimization for a fixed number of training steps.

## A.5 ROLE OF SELECTION CRITERIA

To understand the role of magnitude pruning criteria in the DynSparse training dynamics, we have disentangled the pruning step from the parameter re-allocation step by temporarily replacing the always sparse training algorithm with an alternation between dense and sparse training phases (Peste

et al., 2021). The dense training interval removes the influence from the regrowing selection as all network parameters are periodically activated without preference. We have found that magnitude pruning ("magnitude") outperforms both pruning into a fixed subspace chosen randomly at initialization ("fixed") and a changing random subspace re-drawn each time ("random") as shown in the top half of Table A.16. The strong performance degradation of the random re-drawn sparsity patterns illustrates the importance of the large network parameter in preserving the task performance.

This picture changes if, instead of setting the parameter to zero, we make parameters non-trainable, which avoids the information loss associated with the pruning step. In fact, in this case, we find that randomly selecting the subset of trainable parameters outperforms both the selection of the parameters with the largest magnitude ("magnitude") as well as training a fixed subset of parameters randomly chosen at initialization ("fixed") shown in the bottom part of Table A.16. Our results show that magnitude pruning gives performance advantages as it preserves information. However, the increased exploration coming from random parameter selection would, in the absence of information loss due to pruning, benefit the task performance were it not associated with information loss.

