# OpenReview forum: "Towards Structured Dynamic Sparse Pre-Training of BERT"
_ICLR.cc/2022/Conference — ICLR 2022 Submitted_

### Official Review · Reviewer_5K4c · 2021-10-31

**Correctness:** 3
**Technical Novelty And Significance:** 2
**Empirical Novelty And Significance:** 3
**Recommendation:** 6
**Confidence:** 2

**Main Review:**


Strengths
- The paper shows a simple algorithm (DynSparse) that could help find good sparse
subnetwork in BERT, reduce FLOPS and achieve comparable MLM validation performance.
- The authors did intensive experiments on comparing several different design choices
such as the metric to re-allocate weights, prune the weights, etc.
- The sparse training tool developed is useful for the community.



Weakness

- The novelty of this work is a bit limited. There is no new algorithm proposed to get better results for this sparse training of BERT. The discussions on what could be the reasons behind the design choice are not very convincing to me. For example, random re-allocation has more varieties of network topology explored than RigL, leading to better performance. If this understanding is correct, why would Figure 5 (a) have a convex curve?


- Evaluation of downstream tasks.
Maybe I haven't done a good review of this field, but I am skeptical that MLM validation loss is a good indicator of the quality of pre-trained BERT. Could the authors find references supporting this metric? What if we fine-tune the DynSparse BERT and BERT Large under the similar resource constraint and compare their performance?

- Concern on the scale.
I am not sure if the conclusions still hold when we
scale to larger models.
"DynSparse training can achieve more efficient utilization of FLOPs or network parameters at any scale." It seems a overclaim to me as the paper didn't show the comparison with BERT large.
It would be helpful to label the point of BERT large and its DynSparse counterpart in Figure 3.

- Missing comparison with baseline.
The conclusion from EarlyBert paper is that lottery ticket converges fairly fast and the subnetwork is identified during the early stage of training. This seems to conflict
with the findings in this work that network topology has to be updated sufficiently
to get good results. Could the author elaborate more on this point?
Regardless of always-sparse constraints, how does the quality of subnetwork found
by DynSparse compared to EarlyBERT, in terms of performance in downstream tasks.


**Summary Of The Paper:**

This paper did an extensive investigation of applying dynamic sparse training for BERT.
The main contributions of this work are several empirical observations:
1. The authors empirically show that random re-allocation of weights is better than
gradient-based approach (RigL) for unstructured DynSparse training of Bert.
2. The authors investigated structured DynSparse training and found L1 norm is the
best metric to prune the weights.
3. For a specific range of size of BERT, the DynSparse training shows Pareto improvement (MLM validation loss vs. Non-embedding FLOPS) over the dense BERT.

**Summary Of The Review:**

This work did a comprehensive empirical evaluation on applying DynSpare to BERT,
in both structured and unstructured settings. The novelty of algorithm and theoretical understanding is limited. The empirical results could be more thoroughly conducted, e.g., the scale of BERT, downstream task evaluation.
As an initial step of exploring sparse training of BERT, it provides some promising results. The tool this work developed could benefit the community.

---

> ### Author Response · Authors · 2021-11-19
> **Reply to 5K4c**
>
> Thank you for your thoughtful review and constructive comments.
>
> > The novelty of this work is a bit limited. There is no new algorithm proposed to get better results for this sparse training of BERT.
>
> While the techniques we employ are known, we believe the successful application of these techniques to large model pretraining is novel. This required finding a good learning rate scaling scheme for sparse models, understanding the trade-off between exploration and injected noise with the DynSparse hyperparameters and identifying practically effective components. We also hope that some of the results we found surprising would be a valuable contribution to sparsity research. These include the remarkable similarity between the performance achieved with zero and untrained weights in Figure 1a, and when pruning at different points during training in Figure 1b.
>
> > The discussions on what could be the reasons behind the design choice are not very convincing to me. For example, random re-allocation has more varieties of network topology explored than RigL, leading to better performance. If this understanding is correct, why would Figure 5 (a) have a convex curve?
>
> We consider a trade-off when setting the pruning schedule parameters or applying Rig-L, which is between total explored degrees of freedom (larger is better) and noise injection during training (smaller is better). This guides our interpretation of Figure 5. As we increase the ratio of DOF, since the number of parameters in use at each step is fixed we also necessarily increase the injected noise (of which we have a crude measurement in the "ratio of removed new weights"). Eventually the benefit of increased parameter space exploration is not worth the cost of increased noise.
>
> Our main concern from our experience of Rig-L was from the observed collapse early in training to use only a subset of input features. This collapse implies that it would be more efficiently framed as hidden feature pruning rather than unstructured weight sparsity. Since we were working on the hypothesis that weight sparsity can outperform a dense baseline, we would prefer to avoid this kind of collapse.
>
> > Evaluation of downstream tasks. Maybe I haven't done a good review of this field, but I am skeptical that MLM validation loss is a good indicator of the quality of pre-trained BERT. Could the authors find references supporting this metric? What if we fine-tune the DynSparse BERT and BERT Large under the similar resource constraint and compare their performance?
>
> It is hard to build a general case for correlation between MLM and fine-tuning, which may depend on the nature of the change under test, however:
>
> - ACClip [Zhang et al., 2020] in Table 2 & 3 show ranking agreement between MLM validation loss and SQuAD both across optimiser choice and model scale.
>
> - We note the common use of pretraining loss for autoregressive language models, for example Scaling Laws [Kaplan et al., 2020]. Although autoregressive language modelling is a more “natural” task, we suggest the same approach may be applied to masked language modelling, as long as dataset size is sufficient.
>
> > Concern on the scale. I am not sure if the conclusions still hold when we scale to larger models. "DynSparse training can achieve more efficient utilization of FLOPs or network parameters at any scale." It seems a overclaim to me as the paper didn't show the comparison with BERT large. It would be helpful to label the point of BERT large and its DynSparse counterpart in Figure 3.
>
> We agree that this is an important question to ask and cannot answer definitively beyond the scales we were able to test using the available resources. We have some confidence from the fact that DynSparse maintains a consistent advantage over the baseline family over a wide range of scales, with a similar slope.
>
> > Missing comparison with baseline. The conclusion from EarlyBert paper is that lottery ticket converges fairly fast and the subnetwork is identified during the early stage of training. This seems to conflict with the findings in this work that network topology has to be updated sufficiently to get good results. Could the author elaborate more on this point? Regardless of always-sparse constraints, how does the quality of subnetwork found by DynSparse compared to EarlyBERT, in terms of performance in downstream tasks.
>
> This would be an interesting comparison to make. EarlyBERT operates in a quite different regime of whole-feature pruning using a relatively low sparsity ratio (about 30% for their pretraining experiments). Our hypothesis regarding the difference would be that the amount of training time needed for exploration would increase with always-sparse training and with sparsity ratio. The two techniques also raise different issues regarding compute. High sparsity ratio for EarlyBERT results in memory pressure during the first training phase, while it results in sparse computation issues during DynSparse training.

---

### Official Review · Reviewer_idrP · 2021-11-02

**Correctness:** 3
**Technical Novelty And Significance:** 1
**Empirical Novelty And Significance:** 2
**Recommendation:** 3
**Confidence:** 4

**Main Review:**

Sparse training for neural networks, especially very large transformer-based architectures, is a very  interesting and timely topic when discussing model efficiency. However, the presented paper does not provide much novelty or method advancement. The authors simply apply an already existing method to one type of network. While parameter-studies in general can be of interest, yielding new insights into the training procedure, my main concern is that the studies presented in this paper do not follow a  clear research question, but seem rather random. More importantly, no generalized “lessons-learned” from these studies are drawn, and the translation towards other network architectures is not discussed. As such, the paper lacks significant scientific impact.

Some further comments:
- It is not explained, why the algorithm is adapted to a block-sparse structure. What is the benefit of this, why is it needed?
- All results are reported in terms of reduced FLOPS, but actual training time (wall-clock time) is not considered. Especially on hardware specialized for dense/block structures (GPUs), accessing individual elements in a sparse pattern can, while requiring less operations, result in comparable compute times as their dense counterparts. Moreover, sub optimal usage of the device can thus negate the effects on efficiency, proclaimed by the sparse computation
- In Figure 2, the caption does not match or explain the Figure. How does this Figure illustrate the pruning and re-allocation step? What do the masks and the colours stand for?
- Equation 3: Where does this come from? This seems very empirical. What is the reasoning behind this?
- No details on the utilized hardware and training setting are given

**Summary Of The Paper:**

The paper studies the feasibility to use the dynamic sparsity (DynSparse) method of continuously pruning and re-allocating network weights during training for large unsupervised (self-supervised) language models, namely BERT. Towards this end, the authors present several parameter studies (w.r.t. hyperparameters) of the original BERT training procedure (Mask Language Modeling and Next Sentence Prediction) to assess the Pareto-curve of training BERT with sparse vs dense.

**Summary Of The Review:**

My main concern with this paper is that it is merely a parameter study of an existing method being applied to an existing architecture. While these kind of studies can still be of high scientific value, they need to provide new insight into the method or the architecture and yield some translation/generalization in terms of "lessons learned" for other applications in order to do so. I do not see this to be the case here.

---

> ### Author Response · Authors · 2021-11-19
> **Reply to idrP**
>
> Thank you for your careful review and helpful comments.
>
> > While parameter-studies in general can be of interest, yielding new insights into the training procedure, my main concern is that the studies presented in this paper do not follow a clear research question, but seem rather random. More importantly, no generalized “lessons-learned” from these studies are drawn, and the translation towards other network architectures is not discussed. As such, the paper lacks significant scientific impact.
>
> Our goal in this work was to show that DynSparse training could outperform dense training for large NLP model pretraining. Although there is a reasonably large body of work around sparse training, we found that this was not a straight-forward activity. The details of sparsity ratio, training hyperparameters, learning rates and block sparse pruning metric were critical to achieve good performance. While we agree that we could have done more to draw together our findings clearly, we would argue that these crucial details for applying even simple dynamic sparse training algorithms have value to the community. We would also like to encourage further application of sparse training methods to the challenging regime of large model pretraining over a large dataset, trading off training budget against task performance.
>
> > It is not explained, why the algorithm is adapted to a block-sparse structure. What is the benefit of this, why is it needed?
>
> We were motivated to try block sparsity as one example of structured sparsity, since it allows for more efficient execution on modern highly parallel hardware. This arises because low arithmetic intensity, structure overhead and work imbalance can all harm utilization. Block sparsity can help reduce structure overhead and work imbalance. But this is something we should make more explicit - thank you for this feedback.
>
> > All results are reported in terms of reduced FLOPS, but actual training time (wall-clock time) is not considered. Especially on hardware specialized for dense/block structures (GPUs), accessing individual elements in a sparse pattern can, while requiring less operations, result in comparable compute times as their dense counterparts. Moreover, sub optimal usage of the device can thus negate the effects on efficiency, proclaimed by the sparse computation
>
> This is a key observation, and we believe this is a critical area for the sparsity field. We acknowledge that the training of these models is not wall-clock efficient on many current hardware platforms, however we believe that sufficient proof of the theoretical efficiency of such models would be the foundation for building a model that can be effectively implemented on suitable hardware. We propose a rough separation of the problem in Appendix A.3, where the additional overhead of sparse computation can be summarized as a modified task performance threshold. Advances in hardware could help reduce the extra cost, while advances in training procedure could improve the theoretical gains, both closing the gap. We hope and expect to see advances against both algorithmic and computational challenges in coming years.
>
> > In Figure 2, the caption does not match or explain the Figure. How does this Figure illustrate the pruning and re-allocation step? What do the masks and the colours stand for?
>
> Apologies for insufficient clarity here. The legend is subtle - colours (orange/red/green) match the description below. The bottom part of the figure illustrates a single pruning and re-allocation step, and the chart across the top illustrates the effect of repeated pruning and re-allocation steps on parameter space exploration. We will clarify this in the final manuscript.
>
> > Equation 3: Where does this come from? This seems very empirical. What is the reasoning behind this?
>
> This is indeed an empirical fit, obtained from the grid search of Appendix A.4. As explained in section 2.3, we found that this equation generalized from static to DynSparse and to block DynSparse. While we would like to better understand the effect of sparsity on training dynamics and therefore learning rate, we were forced to leave this to further work.
>
> > No details on the utilized hardware and training setting are given
>
> We detail the hardware platform and software library used to provide sparse operations in section 4 under "Compute-efficient sparse training". Since we did not perform an independently optimized comparison of the dense baseline and DynSparse on training time, we omitted details that did not seem relevant to the included training FLOPs comparison. We are of course happy to elaborate on our setup - are there particular areas that would be of interest?

---

### Official Review · Reviewer_KM7E · 2021-11-03

**Correctness:** 3
**Technical Novelty And Significance:** 2
**Empirical Novelty And Significance:** 2
**Recommendation:** 3
**Confidence:** 4

**Main Review:**

Strengths:
The motivation of using sparse training for BERT language modeling is clear and persuasive.
Extensive experiments and technique details are provided to support claims and demonstrate the effectiveness of the proposed method.

Weaknesses:
It will be better if the author provides more details on the difference between zero and untrained in computation or the time and space complexity analysis of these two methods.
The author mentioned in the part of the contribution that the structured DynSparse training of BERT without structured regularization gives performance gains compared to dense BERT baseline. Structured regularization is used in the block-sparse DynSparse algorithm. Are these two in contradict with each other?
It will be better if the author gives more description of the techniques in the contribution in the part of the methodology.
The author mentioned that the collapse could be mitigated by reducing the influence of activations during training updates. It will help if the author provides more details about how the activations are reduced and how many are reduced compared to the dense baseline.


**Summary Of The Paper:**

Overall, this paper proposed a dynamic sparse pre-training method for BERT language modeling which showed better performance in the number of FLOPs when compared to statically sparse and dense models across a large scale of network sizes.

To be more specific, the proposed DynSparse has four contributions: 1) using random parameter re-allocation to alleviate network parameters collapse; 2) is scalable and FLOPs efficient; 3) showed effectiveness on structured training of BERT 4) structured regularization free while achieving Pareto improvements.

**Summary Of The Review:**

Extensive experiments are conducted to demonstrate the effectiveness of the proposed method. However, I am not fully persuaded by the methodology. More details like complexity analysis and latency reduction experiments could be given.

---

> ### Author Response · Authors · 2021-11-19
> **Reply to KM7E**
>
> Thank you for your review and good questions.
>
> > It will be better if the author provides more details on the difference between zero and untrained in computation or the time and space complexity analysis of these two methods.
>
> We agree that it is interesting to consider the differences between zero and untrained weights. Broadly, untrained weights can save memory and computation in the backward pass and optimizer update, while zero weights also save memory and computation in the forward pass. For this reason, we find zero weights to be preferable, and this was the focus of the investigation. However, we were surprised to see that untrained weights perform very similarly. This gave us confidence that zero weights were not inherently limiting performance (e.g. by disrupting gradient flow).
>
> > The author mentioned in the part of the contribution that the structured DynSparse training of BERT without structured regularization gives performance gains compared to dense BERT baseline. Structured regularization is used in the block-sparse DynSparse algorithm. Are these two in contradict with each other? It will be better if the author gives more description of the techniques in the contribution in the part of the methodology.
>
> Although we described and experimented with Group LASSO as an example of structured regularization, we did not find any improvement compared with properly tuned weight decay, so do not recommend its use in structured DynSparse training. We did, however, find that the decision function that selects groups to be pruned was significant. Is there a particular section where we suggest that group LASSO should be used? We would be keen to make any updates required to clarify our findings here, since they were surprising to us.
>
> > The author mentioned that the collapse could be mitigated by reducing the influence of activations during training updates. It will help if the author provides more details about how the activations are reduced and how many are reduced compared to the dense baseline.
>
> This is an area that would merit further work in trying to understand the behaviour of algorithms such as Rig-L over training. Referring to Figure 11, we observe that the strong horizontal banding of the weight matrix shows that all output weights corresponding to a subset of input features are retained, and most others are lost. Considering the gradient equation for a fully connected layer, this could easily arise from high magnitude input activations. Techniques such as DynSparse do not have such a strong connection between such axes of potential collapse and the pruning decision. It would be interesting to see other attempts at mitigating collapse in Rig-L.
>
> > Extensive experiments are conducted to demonstrate the effectiveness of the proposed method. However, I am not fully persuaded by the methodology. More details like complexity analysis and latency reduction experiments could be given.
>
> We agree that theoretical performance gains on an abstract cost metric such as FLOPs is only one part of a practical DynSparse training system. We propose a rough separation of concerns in Appendix A.3, where the additional overhead of sparse computation can be summarized as a modified task performance threshold. Advances in hardware could help reduce the extra cost, while advances in training procedure could improve the theoretical gains, both closing the gap. While our work here focused on machine learning algorithms for inducing sparsity, we hope and expect to see advances against both algorithmic and computational challenges in coming years.

---

### Official Review · Reviewer_KoFV · 2021-11-04

**Correctness:** 2
**Technical Novelty And Significance:** 2
**Empirical Novelty And Significance:** 3
**Recommendation:** 5
**Confidence:** 4

**Main Review:**

**Positives**

* Reducing the training cost for language models like BERT is an important problem.

* This work covers both *structured* and *unstructured* sparse training for BERT. The proposed block-sparse DynSparse can improve hardware utilization.

* This work suggests minimum efficiency requirements for a time-to-train win for DynSparse over the conventional dense training.


**Concerns/Questions**

* This work in its current form lacks any quantitative comparison against other DynSparse work.  The authors argue for random re-allocation as a means to increase DOF. However, RigL [Evci et al. 2019] suggests that their gradient-based re-allocation outperforms SET, which also adopts random re-allocation, for both vision and NLP tasks (with RNNs, though). Is there any ground for random re-allocation to work better with BERT, or is it simply anecdotal? How would RigL perform with BERT?

* There is no evaluation of end-to-end time ("time-to-train"). Since the authors take into account the utilization in real accelerators by introducing block sparsity, I am wondering how much the proposed technique work reduce the end-to-end training time compared to the baseline dense training.

* Along the line of the previous one, the actual performance gains over dense training may be limited on commodity GPUs. According to Nvidia, cuSparse manages to deliver just about 50% of dense GEMM when using 32x32 blocks [R1]. Depending on the sparsity, the model may fall short of the minimum efficiency requirement of block sparse training to outperform dense training for BERT.

* Evaluation seems a bit too narrow to convince the generality of the proposed technique due to the usage of just one model, fixed sequence length, and lack of ablation study and sensitivity analysis. Can the proposed technique be effective to other models than BERT?  The technique seems quite generic and I am wondering how effective it would be to other models. Is there anything specific to BERT?

[R1] https://developer.nvidia.com/blog/accelerating-matrix-multiplication-with-block-sparse-format-and-nvidia-tensor-cores/


**Summary Of The Paper:**

This paper advocates a straightforward, dynamic sparse (DynSparse) pre-training approach for BERT for more efficient FLOP utilization while maintaining comparable accuracy.  This paper utilizes random reallocation of weights (instead of gradient-based reallocation) to achieve higher total explored degrees of freedom (DOF). This work also proposes a block-sparse DynSparse scheme for higher effective utilization of FLOPS on real hardware accelerators (like GPUs and IPUs).

**Summary Of The Review:**

This paper adapts DynSparse training to BERT to demonstrate its effectiveness. However, evaluation is somewhat narrow and it's not well justified why random re-allocation of weights works better than gradient-based re-allocation for BERT. Also, the performance gains in terms of end-to-end training time on the commodity accelerator like GPUs are still questionable.

---

> ### Author Response · Authors · 2021-11-19
> **Reply to KoFV**
>
> Thank you for your review, questions and constructive feedback.
>
> > This work in its current form lacks any quantitative comparison against other DynSparse work. The authors argue for random re-allocation as a means to increase DOF. However, RigL [Evci et al. 2019] suggests that their gradient-based re-allocation outperforms SET, which also adopts random re-allocation, for both vision and NLP tasks (with RNNs, though). Is there any ground for random re-allocation to work better with BERT, or is it simply anecdotal? How would RigL perform with BERT?
>
> We did try Rig-L, but did not manage to exceed the dense baseline. In a preliminary experiment on BERT-Medium with sparsity ratio 0.9 we achieved eval loss 2.76 for Rig-L versus 2.72 for random reallocation. Note that this is not comparable to our results from the paper, due to different hyperparameter settings. While we cannot rule out good performance of properly tuned Rig-L, we found random reallocation made it easier to achieve FLOP-efficient training.
>
> We suspected this could be due to the "collapse" problem demonstrated in Figure 11. This behaviour makes some intuitive sense if there is positive feedback between weight pruning and backpropagated gradient scale. The Rig-L authors note that the technique compared favourably with pruning in vision tasks, but unfavourably for WikiText-103 character language modelling (although SET performed even worse). We have not been able to tie the collapse problem to NLP over Vision applications, however, and believe a full understanding of task-specific behaviour would be an interesting area for further work.
>
> > There is no evaluation of end-to-end time ("time-to-train"). Since the authors take into account the utilization in real accelerators by introducing block sparsity, I am wondering how much the proposed technique work reduce the end-to-end training time compared to the baseline dense training.
> >
> > Along the line of the previous one, the actual performance gains over dense training may be limited on commodity GPUs. According to Nvidia, cuSparse manages to deliver just about 50% of dense GEMM when using 32x32 blocks [R1]. Depending on the sparsity, the model may fall short of the minimum efficiency requirement of block sparse training to outperform dense training for BERT.
>
> This is a key observation, and we believe this is a critical area for the sparsity field. We acknowledge that the training of these models is not wall-clock efficient on many current hardware platforms, however we believe that sufficient proof of the theoretical efficiency of such models would be the foundation for building a model that can be effectively implemented on suitable hardware. We propose a rough separation of the problem in Appendix A.3, where the additional overhead of sparse computation can be summarized as a modified task performance threshold. Advances in hardware could help reduce the extra cost, while advances in training procedure could improve the theoretical gains, both closing the gap. We hope and expect to see advances against both algorithmic and computational challenges in coming years.
>
> > Evaluation seems a bit too narrow to convince the generality of the proposed technique due to the usage of just one model, fixed sequence length, and lack of ablation study and sensitivity analysis. Can the proposed technique be effective to other models than BERT? The technique seems quite generic and I am wondering how effective it would be to other models. Is there anything specific to BERT?
>
> This is a valid criticism - we do not provide direct evidence to support the generality of the technique. However, limiting ourselves in these respects did have some advantages: we have been able to evaluate models for our single problem at reasonable scale and explore sweeps of hyperparameters such as sparsity ratio and pruning frequency. This has given us general technique for considering these parameters that we would expect to transfer well to other problems. For example, Figures 4 and 5 help to ground decisions regarding sparsity ratio, pruning ratio and update frequency. We would expect such general findings to be independent of BERT, but the optimal values of these hyperparameters and scale of DynSparse advantage to be task dependent.

---

### Official Review · Reviewer_vEBH · 2021-11-08

**Correctness:** 3
**Technical Novelty And Significance:** 2
**Empirical Novelty And Significance:** 2
**Recommendation:** 3
**Confidence:** 3

**Details Of Ethics Concerns:**

No ethics concerns

**Main Review:**

The authors propose to dynamically prune the transformer to improve the efficiency of BERT pre-training. To maintain the computational regularization, block sparsity is used in this work. The main technique of this paper is to add group LASSO regularization to induced sparsity and prune the network depending on the magnitude of the weights. The proposed method is simple yet effective. The resulting sparse pre-training method is nearly twice as efficient as the original BERT pre-training method. However, the novelty of this paper is limited because the proposed sparse pruning technique has been widely used to train convolutional neural networks. Moreover, it remains unclear whether the proposed method is tailored specifically for pre-training of sequence models. My additional concerns are listed below.

1) Comparison with other dynamic approaches: As discussed in this paper, other lines of research exist to explore sparse transformer architectures [1, 2]. However, the proposed method is not compare with these prior works in the paper.

2) Wall-clock evaluation: No wall-clock evaluation is provided. It is not clear whether the proposed method can provide real speedup on accelerator chips such as google TPU.

3) Finetuning results: The authors only show the MLM loss after pre-training as the quality metric. However, it is also important to show the performance of models after fine-tuning on GLUE / SQuAD datasets.

[1] Accelerating Training of Transformer-Based Language Models with Progressive Layer Dropping
[2] Reducing Transformer Depth on Demand with Structured Dropout

**Summary Of The Paper:**

The authors propose a dynamic sparse pretraining method to reduce the computational cost of BERT pretraining.

**Summary Of The Review:**

The novelty of the proposed method is limited and the empirical evidence is not strong enough to justify the method.

---

> ### Author Response · Authors · 2021-11-19
> **Reply to vEBH**
>
> Thank you for your review and helpful comments on our work.
>
> You mentioned that the main technique of the paper was to add group LASSO regularization, but in fact we found simple elementwise weight decay to outperform group LASSO. Instead, we claim novelty in applying existing sparse training techniques to a more challenging task than has been previously shown – improving the training efficiency of large language models in pretraining.
>
> > However, the novelty of this paper is limited because the proposed sparse pruning technique has been widely used to train convolutional neural networks.
>
> While the techniques we employ are all known, we believe the successful application of these techniques to large model pretraining is novel. This required finding a good learning rate scaling scheme for sparse models, understanding the trade-off between exploration and injected noise with the DynSparse hyperparameters and identifying the practically effective components from existing literature. We would also suggest that the baseline of a Transformer encoder for BERT is considerably stronger than that of ResNet-50 for CIFAR or ImageNet. This raises the question of whether dynamic sparse training has anything to offer to models such as BERT, which we sought to address.
>
> > Moreover, it remains unclear whether the proposed method is tailored specifically for pre-training of sequence models.
>
> We would expect the parameters used to be specific to sequence model pretraining. However, we would propose retaining the general approach of increasing the learning rate with sparsity and selecting pruning interval and ratio to maximise explored network parameters DOF, while minimising noise injection.
>
> > Comparison with other dynamic approaches: As discussed in this paper, other lines of research exist to explore sparse transformer architectures [1, 2]. However, the proposed method is not compare with these prior works in the paper.
>
> We have indeed tried some alternative dynamic approaches. While it would be interesting to compare against the techniques you mentioned, we concentrated our efforts on more fine-grained sparsity techniques such as Rig-L [Evci et al. 2019] since they are most similar to DynSparse. However, given failure to even achieve efficiency versus the dense baseline, we cannot be confident that we are fairly representing the technique (for example, it may require additional tuning). For this reason, we were reluctant to include such results.
>
> In a preliminary experiment on BERT-Medium with sparsity ratio 0.9 we achieved eval loss 2.76 for Rig-L versus 2.72 for random reallocation. Note that this is not comparable to our results from the paper, due to different hyperparameter settings. While we cannot rule out good performance of properly tuned Rig-L, we found random reallocation made it easier to achieve FLOP-efficient training.
>
> > Wall-clock evaluation: No wall-clock evaluation is provided. It is not clear whether the proposed method can provide real speedup on accelerator chips such as google TPU.
>
> This is a key observation, and we believe this is a critical area for the sparsity field. We acknowledge that the training of these models is not wall-clock efficient on many current hardware platforms, however we believe that sufficient proof of the theoretical efficiency of such models would be the foundation for building a model that can be effectively implemented on suitable hardware. We propose a rough separation of the problem in Appendix A.3, where the additional overhead of sparse computation can be summarized as a modified task performance threshold. Advances in hardware could help reduce the extra cost, while advances in training procedure could improve the theoretical gains, both closing the gap. We hope and expect to see advances against both algorithmic and computational challenges in coming years.
>
> > Finetuning results: The authors only show the MLM loss after pre-training as the quality metric. However, it is also important to show the performance of models after fine-tuning on GLUE / SQuAD datasets.
>
> We accept this criticism, an unfortunate omission that is due to current limitations of our training setup, that we are working on resolving. We are reassured by some evidence for correlation between MLM and fine-tuning. Zhang et al. [1] in Table 2 & 3, show ranking agreement between MLM validation loss and SQuAD both across optimiser choice and model scale. But as you suggest, we would be interested to evaluate fine-tuning performance to confirm that this is the case. These experiments would raise the additional question of whether we should keep the sparsity pattern fixed after pretraining, or continue with a DynSparse scheme during fine-tuning.
>
> ---
>
> [1] Zhang, J., Karimireddy, S.P., Veit, A., Kim, S., Reddi, S.J., Kumar, S. and Sra, S., 2019. Why are adaptive methods good for attention models?. arXiv preprint arXiv:1912.03194.

---

### Decision · Program_Chairs · 2022-01-20

**Decision:**

Reject

**Comment:**

All of the reviewers believe the paper should not be accepted, and I concur with their recommendation for the reasons they mention.

Four of the reviewers (vEBH, idrP, KoFV, 5k4c) believe the technique proposed in this paper is not particularly novel. Rather, the novelty is that it is being used on a BERT model rather than the computer vision models that are typically the starting point for pruning work. They also argue that the paper was not particularly thorough in its comparison to other pruning techniques (specifically dynamic sparsity techniques), which is essential for pruning work given how crowded and noisy the space is. Finally, they rightfully note that the paper does not look at the real-world speedups attainable on conventional hardware (GPUs and TPUs), the latter of which has no support for sparsity and the former of which (NVIDIA Ampere) has limited support for specific kinds of sparsity and especially limited support for sparse training.

The reviewers also raised several more specific methodological issues with evaluation (e.g., using the MLM loss rather than fine-tuning as a basis for evaluation), but the above concerns alone were enough to convince me that the paper does not merit acceptance at this time.